# Synergistic Effects of Korean Mistletoe and Apple Peel Extracts on Muscle Strength and Endurance

**DOI:** 10.3390/nu16193255

**Published:** 2024-09-26

**Authors:** Youn-Goo Kang, Joonhyuk Kwon, Soonjun Kwon, Ah-Ram Kim

**Affiliations:** 1Department of Advanced Convergence, Handong Global University, Pohang 37554, Republic of Korea; ygkang@handong.ac.kr; 2School of Life Science, Handong Global University, Pohang 37554, Republic of Korea

**Keywords:** mistletoe, apple peel, muscle strength, muscle endurance, synergistic effect

## Abstract

Muscular strength and endurance are vital for physical fitness. While mistletoe extract has shown efficacy in significantly increasing muscle strength and endurance, its accessibility is limited. This study explores combining mistletoe and apple peel extracts as an effective muscle health supplement. Analyses of histology, RNA, and protein in the combined extract-treated mouse group demonstrated significant enhancements in muscle strength and endurance, evidenced by larger muscle fibers, improved mitochondrial function, and a higher ratio of type I and IIa muscle fibers. Combining half doses of each extract resulted in greater improvements than using each extract separately, indicating a synergistic effect. Pathway analysis suggests that the observed synergy arises from complementary mechanisms, with a mistletoe extract-induced decrease in myostatin (MSTN) and an apple peel extract-induced increase in IGF1, leading to a sharp rise in AKT, S6K, and MuRF1, which promote myogenesis, along with a significant increase in PGC-1α, TFAM, and MEF2C, which are critical for mitochondrial biogenesis. This research provides practical insights into developing cost-effective, natural supplements to enhance muscle performance and endurance, with potential applications in athletic performance, improving muscle growth and endurance in children, and addressing age-related muscle decline.

## 1. Introduction

Physical fitness is a state of health and well-being that involves the ability to perform daily activities with vigor and without excessive fatigue [1]. It encompasses muscular strength, muscular endurance, cardiovascular endurance, flexibility, and body composition [2,3]. Muscular strength is the amount of force a muscle can exert against a resistance in a single effort [4]. Enhancing muscular strength improves basic physical abilities such as climbing, lifting, pushing, and pulling, making it easier to perform various daily activities [5]. Meanwhile, muscular endurance is the ability of the human body to sustain continuous activity over extended periods of time [6], and enhancing muscular endurance contributes to providing sustained energy for everyday activities and improving resistance to muscle fatigue [7]. Both muscular strength and muscular endurance are essential for maintaining healthy fitness levels and a high quality of life.

Skeletal muscle mass decreases at a rate of approximately 3–8% per decade after the age of 30, and the rate of loss accelerates after the age of 60 [8]. Therefore, maintaining skeletal muscle through physical activity is crucial. However, technological advancements and automation in modern society are leading to a gradual decrease in physical activity [9]. According to a report published by the World Health Organization (WHO) in 2016, 28% of adults aged 18 and over worldwide failed to meet the weekly activity guidelines [10]. The reduction in physical activity can lead to a decrease in muscle mass and strength, which in turn can lower metabolic rates, reduce physical capabilities, and increase the risk of certain chronic diseases [11,12,13]. Research indicates that, in Europe, approximately 676,000 deaths each year are linked to inadequate physical activity, which is slightly more than double the number of deaths associated with obesity, estimated at 337,000 [14].

Throughout history, humans have sought to enhance physical strength and endurance using plant based natural products [15,16,17]. Plant compounds like curcumin, resveratrol, and catechin have shown promise as potent enhancers of muscle physiology [18]. Resveratrol, in particular, exhibits benefits in promoting protein synthesis, reducing protein degradation, and alleviating muscle fiber atrophy [19,20]. A rodent study administering resveratrol at a dose of 400 mg/kg per day showed a significant reduction in muscle fiber atrophy [21]. Mistletoe extract is also notable for its potential to enhance muscle strength and endurance. A study comparing the effects of mistletoe extracts versus resveratrol on exercise performance in mice found that mistletoe extracts resulted in significantly longer swimming times (over 500 min) compared to resveratrol (approximately 400 min) and the negative control (approximately 190 min) [22]. This highlights mistletoe extract’s superior efficacy in boosting muscular endurance. Subsequent studies have also reported the effects of mistletoe in enhancing muscle strength and endurance [23,24,25]. The diverse functionality of mistletoe extract is presumed to be due to its various bioactive constituents, including lectins, viscotoxins, polyphenols (flavonoids, phenolic acids, sterols, and lignans), triterpenoids, phenylpropanoids, alkaloids, and fatty acids [26]. However, due to its parasitic nature, the chemical composition and functional properties of mistletoe extracts exhibit significant variability depending on the host tree species [27]. Additionally, the plant’s tendency to grow in the upper tree canopy complicates harvesting, contributing to its high cost and limiting its utilization in health supplement development [28]. Moreover, high dose intake of mistletoe extracts can induce a range of adverse effects, including fever, fatigue, and potentially life-threatening complications such as anaphylactic shock or liver damage [29,30,31].

To address the limitations of mistletoe, this study explored natural products that could be combined with mistletoe extract to complement its limitations while enhancing its effectiveness. Apples, like mistletoe, contain high levels of polyphenolic compounds, but the types of these compounds differ significantly [32,33]. While mistletoe is rich in lignans, for example, apples are rich in quercetin and its related glycosides, which belong to flavonoids, a major group of polyphenols [34]. Consuming apples or their flavonoid components, including quercetin, has been shown to improve muscle strength and endurance [35,36,37].

We hypothesized that combining mistletoe with apple extracts could potentially enhance muscle strength and endurance due to the complementary effects of their different compositions, including polyphenolic compounds. It is also noteworthy that many beneficial nutrients are predominantly located in apple peels, which are often discarded during apple processing [34,38]. The apple peels generated during the processing of washed apples are not only easily accessible and safe to eat, but also very affordable, making them highly promising as functional food additives [39]. The primary objective of this study is to assess the potential of apple peel extract, alone or in combination with mistletoe, to complement and enhance the efficacy of mistletoe in improving muscle strength and endurance, as well as to elucidate the mechanisms of action.

## 2. Materials and Methods

### 2.1. Natural Products and Chemical Materials

The Korean mistletoe was collected from oak trees in Gangwon-do, Republic of Korea, at one or two years of age. Fuji apples, cultivated in Pohang, Republic of Korea, were obtained from the Pohang agriculture wholesale market (Pohang, Gyeongsangbuk-do, Republic of Korea). Ethanol for apple peel extraction was obtained from DAEJUNG Chemicals & Metals (Siheung, Gyeonggi-do, Republic of Korea). Folin–Ciocalteu phenol reagent, used for total polyphenol content measurement, was obtained from Sigma-Aldrich (St. Louis, MO, USA) and Na_2_CO_3_ and gallic acid were obtained from DAEJUNG Chemicals & Metals. Aluminum nitrate, potassium acetate, and quercetin, which were used for flavonoid content analysis, and creatine monohydrate, which served as a positive control in animal experiments, were obtained from DAEJUNG Chemicals & Metals.

### 2.2. Mistletoe and Apple Peel Extract Preparation

The mistletoe extract was prepared according to the method described by Jung et al. [24]. In brief, the leaves, stems, and fruits of Korean mistletoe were collected from the distal ends of branches, washed with distilled water (DW), and dried. The dried mistletoe components were mashed and ground in a 1:10 (*w*/*v*) DW water solution for 30 s. After washing, the homogenate was further blended with a blender for 2 min and then stirred continuously for 16 h at 4 °C. It was then centrifuged at 6500× *g* for 30 min at 4 °C. The clear supernatant was sequentially filtered through 0.9 µm and 0.45 µm filters. The resulting filtrate, now the mistletoe extract, was subsequently freeze-dried.

Apple peel extract was prepared according to the following steps. Fresh apples were peeled, washed with DW, and dried using a food dehydrator (OCOO, Boryeong, Chungcheonnam-do, Republic of Korea) at 40 °C for 48 h. The dried apple peels were blended in a blender for 5 min. The blended apple peel powder was mixed with 60% ethanol in a 1:10 ratio and extracted for 12 h at room temperature using a magnetic stirrer. After the extraction was completed, the mixture was filtered through a 0.2 µm syringe filter (Sartorius, Göttingen, Land Niedersachsen, Germany) and freeze-dried using a FD8508 (IlshinBioBase Co., Ltd., Dongducheon, Gyeonggi-do, Republic of Korea) for 24 h. The freeze-dried apple peel extract was stored at −80 °C until it was used for the animal experiments. 

### 2.3. Total Polyphenol Measurement

The total polyphenol content in the sample was measured using the Folin–Ciocalteu method [40]. In brief, 40 µL of the sample and 200 µL of Folin–Ciocalteu phenol reagent were mixed in a 96-well plate (Avantor, Radnor Township, PA, USA) and incubated for 5 min at room temperature. Then, 160 µL of 10% Na_2_CO_3_ was added and incubated at room temperature for one hour. Absorbance was measured at 765 nm using a SPECTROstar Nano (BMG LABTECH, Ortenberg, Germany). A standard curve was generated using gallic acid with concentrations ranging from 0 to 100 µg/mL.

### 2.4. Total Flavonoids Measurement

Flavonoids undergo a structural change, turning yellow in color, when exposed to strong alkaline conditions [41]. In brief, 40 µL of the sample was added to each well of a 96-well plate. With the addition of 8 µL of 10% aluminum nitrate, 8 µL of 1 M potassium acetate, and 344 µL of ethanol sequentially, the mixture was incubated at room temperature for 40 min. Absorbance was measured at 415 nm using a SPECTROstar Nano. A standard curve was generated using quercetin, with concentrations ranging from 0 to 100 µg/mL.

### 2.5. Animal Experiments

Four-week-old ICR male mice bought from Saeronbio Inc. (Uiwang, Gyeonggi-do, Republic of Korea) were housed at 23 ± 1 °C and 60 ± 5% humidity, and a 12 h light/dark cycle. After one week of adaptation, the mice were divided into six groups (*n* = 7 per group). Table 1 provides a detailed description of the groups. The rodents were fed Purina Lab Rodent Chow as their diet (Purina, Seongnam, Gyeonggi-do, Republic of Korea). For four weeks, oral administration of creatine, apple peel extract, and the mistletoe extract was performed once a day according to the groups, at doses of 100 mg/kg or 200 mg/kg. After a four-week treatment period, a grip strength test and treadmill test were performed. On the last day of the animal experiment, the mice were sacrificed by CO_2_ gas. Gastrocnemius (GA) muscles were collected and stored at −80 °C until used. All animal experiments were performed in accordance with protocols approved by the Institutional Animal Care and Use Committee of Handong Global University (IACUC no. 20230719-07).

### 2.6. Measurement of Muscle Strength

The grip strength of mice was evaluated by the BIO-GS3 Force Gauge (Bioseb, Pinellas Park, FL, USA), with measurements recorded in units of G force. The instrument provided a readout of the maximum force exerted by a mouse as it was gently pulled downwards. For each individual mouse, five consecutive trials were conducted, disregarding the two lowest readings to eliminate outliers. The final measurement value was determined by calculating the average of the other three measurements.

### 2.7. Measurement of Muscle Endurance

For measuring endurance capacity, the Exer 6M Treadmill (Columbus instruments, Columbus, OH, USA) was used, consisting of an electric shock grid at the rear of the treadmill machine. The current from the shock grid was set to between 0.2 mA to 0.3 mA, giving an uncomfortable shock but no physical harm or injury to the mice. One day prior to the treadmill test, all mice from each group were acclimatized with the settings of 20 m/min for 15 min. After acclimatization, the test was initiated at 12 m/min and the speed was increased by adding 3 m/min every 3 min. No further speed increment was done after reaching 21 m/min. Exercise performance for each mouse was measured when the mice reached exhaustion, showing inability to run for 10 s.

### 2.8. Histological Analysis

The parts of gastrocnemius muscle were fixed in 10% *v*/*v* formalin/PBS for histology analysis. They were embedded in paraffin for staining with haematoxylin and eosin. Images were obtained by ZEISS Axio Imager 2 (Carl Zeiss Co., Ltd., Oberkochen Baden-Württemberg, Germany) at a magnification of 200×. The cross-sectional area (CSA) analysis was conducted using ImageJ software (version 1.53t) with the “Cross sectional Analyzer” plug-in, following the developer’s instructions [42].

### 2.9. Real-Time RT PCR

The total RNA was extracted from GA muscle using the TRIzol Reagent (Invitrogen, WALTHAM, MA, USA). cDNA was synthesized with a SuperiorScript III cDNA Synthesis kit (Enzynomics, Inc., Daejeon, Republic of Korea). Quantitative PCR (qPCR) was conducted by Applied Biosystem StepOnePlus^TM^ Real-Time PCR system (Applied Biosystems, Waltham, MA, USA) with a TOPreal^TM^ qPCR 2X PreMIX (SYBR Green with high ROX) kit (Enzynomics, Inc.). All primers used for the qPCR were synthesized by Macrogen (Seoul, Republic of Korea), and their sequences are provided in Appendix A. The results are presented as means ± S.D, normalized to GAPDH expression using the ∆∆Ct method, in which the NC group was used as the reference group.

### 2.10. Western Blotting

GA tissues were homogenized in the PRO-PREP^TM^ Protein extraction solution (iNtRON Biotechnology, Seongnam, Gyeonggi-do, Republic of Korea) containing an Xpert Duo inhibitor Cocktail Solution (GenDEPOT, Katy, TX, USA). Equal amounts of protein were separated by 10% sodium dodecyl sulfate-polyacrylamide gel electrophoresis (SDS-PAGE) and transferred to polyvinylidene difluoride (PVDF) membranes. Membranes were blocked in a blocking solution containing 5% skim milk for 1 h at room temperature and incubated overnight at 4 °C with the following primary antibodies: anti-AMPKα (#2532; Cell Signaling Technology (CST), Danvers, MA, USA), anti-Phospho-AMPKα (Thr172) (#2535; CST), anti-AKT (#9272; CST), anti-Phospho-AKT (Ser473) (#4058; CST), and anti-GAPDH (14C10) (#2118; CST). The membrane was washed three times for 1 min and two times for 5 min with tris-buffered saline with Tween-20 (TBST buffer), followed by incubation with HRP-conjugated anti-rabbit IgG as secondary antibodies (#7074; CST). Immunoblots were visualized by SuperSignal^TM^ West Femto Maximum Sensitivity Substrate (Thermo Scientific^TM^, Waltham, MA, USA) and densitometric analyses were done using ImageJ software (version 1.53t).

### 2.11. Immunohistochemistry

Tissues were fixed in 10% *v*/*v* formalin/PBS, followed by paraffin embedding. Sections were cut at 4 µm of thickness and affixed to coated slides. After heat treatment, immunohistochemistry (IHC) was conducted. Briefly, samples underwent microwave pre-treatment and were then treated with peroxidase blocking solution (Dako Denmark A/S, Glostrup, Denmark) at room temperature for 10 min to minimize nonspecific background staining. Serum-free protein blocking buffer (Dako Denmark A/S) was applied at room temperature for 1 h, and then samples were incubated overnight at 4 °C with primary antibodies: MyHC1 (#BA-F8; Developmental Studies Hybridoma Bank (DSHB), Iowa City, IA, USA), MyHC2A (#SC-71; DSHB), and MyHC2B (BF-F3; DSHB). After primary antibody staining, the secondary antibody was applied using EnVision+ Single Reagent (HRP, Mouse) (Dako Denmark A/S). Nuclear counterstaining was performed with Mayer’s Hematoxylin (Dako Denmark A/S), and IHC samples were completed with a glass cover slip. Images were obtained by ZEISS Axio Imager 2 (Carl Zeiss Co., Ltd.) at a magnification of 200× and quantification of Western blot was done using ImageJ software (version 1.53t).

### 2.12. Transmission Electron Microscopy (TEM)

The number of mitochondria in skeletal muscle was measured using transmission electron microscopy (TEM). Samples were fixed with Karnovsky’s fixative (2% paraformaldehyde and 2% glutaraldehyde (EMS) in 0.05 M sodium cacodylate buffer (pH 7.2)) (Electron Microscopy Sciences (EMS)), then post-fixed in 1% osmium tetroxide (EMS) in 0.1 M sodium cacodylate buffer (pH 7.2). After dehydration in a graded alcohol series, the samples were embedded in Embed 812-kit (EMS). Ultrathin sections, 100 nm thick, were cut with a Leica UC7 (Leica microsystems, Wetzlar, Germany), collected on 200-mesh copper grids, double stained with 2.5% uranyl acetate (TED PELLA Inc., Redding, CA, USA) and 2.5% lead citrate (EMS). The specimens were observed using a Talos L120C (Thermo Fisher Scientific^TM^) at 120 kV.

### 2.13. Statistical Analysis

Statistical analyses were performed using Origin2022b (OriginLab., Northampton, MA, USA). The experimental results were presented as means ± S.D and analyzed with a one-way ANOVA with Tukey’s post-hoc test compared with different groups; *p*-value < 0.05 were considered as statistically significant.

## 3. Results

### 3.1. Optimizing the Apple Peel Extraction Method to Maximize Flavonoids and Polyphenol Content

Unlike mistletoe, which has optimized extraction techniques through various studies, apple peels still lack efficient extraction methods [23,24,25]. To effectively extract polyphenols, including flavonoids, which are considered one of the most beneficial compounds in apple peels, the effects of varying ethanol concentrations and extraction time in ethanol were compared. Ethanol is widely recognized as a safe and universal solvent due to its capacity to dissolve both polar and nonpolar compounds, facilitated by its molecular structure that possesses polarity and hydrophobicity [43]. Different ethanol concentrations ranging from 0% to 100% were examined over a fixed duration of 12 h, unveiling the peak yield at 60% ethanol (Figure 1a). In experiments changing the extraction time (0–12 h) at a fixed ethanol concentration (60%), which exhibited the highest efficiency, the optimal extraction time was identified as 12 h (Figure 1b). Based on these findings, apple peel extracts were prepared for further experiments using 60% ethanol for 12 h. The final concentration of polyphenols and flavonoids in the prepared samples was measured (Figure 1c). Neither the negative control group (NC), which received only phosphate buffered saline (PBS), nor the positive control group (PC), which was supplemented with creatine, contained any detectable polyphenols or flavonoids. “A group” refers to the group treated with the apple peel extract. The sample given to the A group exhibited a concentration of 1.08 mg/mL polyphenols and 0.65 mg/mL flavonoids. “M group” refers to the group treated with the mistletoe extract. The sample given to the M group exhibited a concentration of 1.09 mg/mL polyphenols and 0.08 mg/mL flavonoids. The “AML group” refers to the group treated with a combination of apple peel extract and mistletoe extract, each at half the concentration compared to the A group or M group. Consequently, the concentrations of polyphenols (1.15 mg/mL) and flavonoids (0.36 mg/mL) in the AML group were equal to half the sum of the concentrations found in the A group and the M group. Finally, the “AMH group” refers to the group treated with a combination of apple peel extract and mistletoe extract at the same concentration as the individual A group and M group. Consequently, the concentrations of polyphenols (2.21 mg/mL) and flavonoids (0.98 mg/mL) in the AMH group were equal to the sum of the concentrations found in the A group and the M group. 

### 3.2. Synergistic Effect of Combined Mistletoe and Apple Peel Extracts on Muscle Strength and Muscular Endurance

To assess the effects of individual and combined extracts on muscle strength and muscular endurance, mice were orally administered apple peel extract (A group), mistletoe extract (M group), their combination (AML and AMH group), creatine (PC group), or PBS (NC group) for four weeks. During the administration period, it was confirmed that there were no significant differences in calorie intake between the groups (Figure 2d). Following the administration period, muscle strength and endurance were compared between the groups through animal behavior experiments (Figure 2a). 

Muscle strength was evaluated using a grip strength meter. Compared to the NC group, the PC group exhibited a significant increase in muscle strength (Figure 2b). Interestingly, neither the A nor the M groups displayed significant changes in muscle strength compared to the NC group. However, the AML and AMH groups demonstrated a significant enhancement in muscle strength compared to the NC group. Notably, the AML group exhibited a significant increase even when using only half the concentration of each individual extract, suggesting a synergistic effect.

Muscle endurance was evaluated using a treadmill test. Compared to the NC group, neither the PC group nor the A group displayed significant changes in running time (Figure 2c). Conversely, the M group displayed a significant increase in running time compared to the NC group. These findings support previous research [24]. Remarkably, the AML and AMH groups demonstrated significant increases in running time (233% and 258%, respectively) compared to not only the NC group but also the A and M groups.

### 3.3. Combined Mistletoe and Apple Peel Extracts Exerts a Synergistic Effect on Increasing the Size of Muscle Fibers

To analyze the extract’s effect on muscle strength improvement in more detail, hematoxylin and eosin (H&E) staining was performed on GA muscle from each treatment group to evaluate histology and measure myofiber cross-sectional area (CSA). The AMH group had the highest proportion of myofibers within the consistent area in comparison to other groups, while the NC group exhibited the lowest proportion (Figure 3a,b). Analysis of muscle fiber CSA revealed a significant increase in average fiber size in both the A and M groups compared to the NC group (Figure 3c,d). Notably, the AML groups showed an even more significant increase in average CSA compared to the A and M groups (Figure 3c).

To elucidate the molecular mechanisms underlying the observed changes in muscle, gene expression and protein phosphorylation levels were analyzed. The analysis focused on genes related to protein degradation (Forkhead box protein O1: FOXO1, Muscle RING-finger protein-1: MuRF1, F-box protein 32: Atrogin-1, Tumor necrosis factor alpha: TNFα, and Nuclear factor kappa B: NF-κB), myogenesis (Myostatin: MSTN, Myoblast determination protein 1: MyoD), and protein synthesis (Insulin-like growth factor-1: IGF1, Eukaryotic translation initiation factor 4E-binding protein 1: 4EBP1, Ribosomal protein S6 kinase: S6K, and Protein kinase B: AKT). Additionally, the phosphorylation level of AKT at Ser473, a critical marker of protein activation, was observed. In a comparative analysis of the expression levels of five genes associated with protein degradation (FOXO1, MuRF1, Atrogin-1, TNFα, and NF-κB) across different groups, it was observed that the A and M group displayed a significant decrease in the expression of all evaluated genes compared to the NC group, except for TNFα in the A group and Atrogin-1 and FOXO1 in the M group (Figure 3e). The AML and AMH groups exhibited significant suppression in the expression of all five genes compared to the NC group. Notably, in the AML group, the expression of MuRF1 was significantly reduced compared to the A group and the M group (Figure 3e). This suggests that the combination of mistletoe and apple peel extracts has a synergistic effect on mitigating protein degradation. Upon comparing the expression levels of two genes associated with myogenesis (MSTN and MyoD) across different groups, the M group demonstrated a significant decrease in the expression of MSTN, alongside an increase in MyoD expression compared to the NC group (Figure 3f). Both the AML and AMH groups exhibited significant decreases in the expression levels of MSTN and increases in the levels of MyoD relative to the NC group. In the gene expression analysis of four genes associated with protein synthesis (IGF1, 4EBP1, S6K, and AKT), a significant increase in the expression of IGF1, S6K, and AKT and a decrease in 4EBP1 expression were noted in the A group relative to the NC group (Figure 3g,h). The AML and AMH groups demonstrated substantial increases in IGF1, S6K, and AKT expressions and decreases in the 4EBP1 expression relative to the NC group. Notably, the expression of S6K in the AML group was significantly increased compare with those of both the A and M groups, indicating a synergistic effect in increasing protein synthesis.

Moreover, the A groups exhibited a significant elevation in the ratio of phosphorylated AKT to total AKT protein compared to the NC group, indicating enhanced pathway activation (Figure 3i,j). The AML and AMH groups demonstrated a significantly higher phosphorylated AKT/total AKT ratio compared with the A and M groups, suggesting a synergistic enhancement in the Akt-mediated protein synthesis pathway.

### 3.4. Combined Mistletoe and Apple Peel Extracts Exerts a Synergistic Effect on Enhancing Mitochondrial Biogenesis and Mitochondrial Fusion

Animal behavior experiments revealed a synergistic effect of the combined mistletoe and apple peel extracts on increasing muscle endurance capacity (Figure 2c). Given that mitochondrial biogenesis and mitochondrial dynamics are well established as key factors influencing muscular endurance enhancement [44], a comparative analysis of mitochondrial abundance in the GA muscle, utilizing transmission electron microscopy (TEM), revealed a significantly higher number of mitochondria in the M group compared to both the NC and A groups. Notably, the AML group exhibited the highest mitochondrial content amongst all groups, demonstrating a statistically significant difference from the other groups (Figure 4a,b). 

To elucidate the molecular mechanisms underlying the observed changes in muscular endurance, the focus was on genes related to mitochondrial biogenesis (Sirtuin 1: SIRT1, Peroxisome proliferator activated receptor gamma coactivator 1 alpha: PGC-1α, and transcription factor A, mitochondrial: TFAM), mitochondrial fission (dynamin related protein: DRP1, mitochondrial fission 1 protein: FIS1), mitochondrial fusion (mitofusin 1: MFN1, mitofusin 2: MFN2) and AMP activated protein kinase (AMPK) along with AMPK phosphorylation as a marker of its relevant pathway activation. In the comparative analysis of expression levels of three genes implicated in mitochondrial biogenesis (SIRT1, PGC-1α, and TFAM) across various groups, the M group demonstrated a significant increase in the expression of SIRT1 and TFAM relative to the NC group (Figure 4e). The AML and AMH groups demonstrated substantial increases in all three gene expressions relative to the NC group, with PGC-1α expression in the AML group being significantly higher than that in either the A or M groups, indicating a synergistic effect in increasing mitochondrial biogenesis. In the comparative analysis of expression levels of two genes implicated in mitochondrial fission (DRP1 and FIS1) across various groups, the A group and M group demonstrated a significant decrease in the expression of DRP1 and FIS1 relative to the NC group (Figure 4c). The AML and AMH groups demonstrated a substantial decrease in both gene expressions relative to the NC group. In the comparative analysis of expression levels of two genes implicated in mitochondrial fusion (MFN1 and MFN2) across various groups, the A group demonstrated a significant increase in the expression of MFN1 and MFN2 relative to the NC group (Figure 4d). The AML and AMH groups demonstrated substantial increases in both gene expressions relative to the NC group. The M groups exhibited a significant increase in the gene expression of AMPK and elevation in the ratio of phosphorylated AMPK to total AMPK protein compared to the NC group, indicating enhanced pathway activation (Figure 4f–h). The AML and AMH groups demonstrated a significantly higher phosphorylated AMPK/total AMPK ratio than the A and M groups, suggesting a synergistic enhancement in pathway activation.

### 3.5. Synergistic Effect of the Combine Mistletoe and Apple Peel Extracts on Increasing Different Types of Muscle Fibers

Mitochondrial biogenesis function in skeletal muscle is closely linked to the fiber type composition [45]. Fiber types I and 2A, which are known to affect muscular endurance, typically exhibit high mitochondrial density [46]. Our study found that the combination of mistletoe extract and apple peel extract increases the number of mitochondria and the expression of genes related to mitochondrial biogenesis (Figure 4a,b,e). 

Based on these results, immunohistochemistry (IHC) was employed on GA muscle cross-sections to determine the effect of this combination on muscle fiber type composition. The antibodies targeting specific markers for type I (MyHC-1), type IIa (MyHC-2A), and type IIb (MyHC-2B) fibers were utilized (Figure 5a). In the A group, only the area of type IIa fibers exhibited a significant increase compared to the NC group (Figure 5b). Conversely, the M group displayed significant increases in the areas of both type I and type IIa fibers compared to the NC group (Figure 5b). The AML group demonstrated significant increases in the composition of all muscle fiber types compared to the NC group (Figure 5b). Notably, despite the AML group receiving a combination of mistletoe extract and apple peel extract at half the concentration of the A group and M group, respectively, the type I fibers significantly increased compared to the A group. Additionally, the type IIa and type IIb fibers showed significant increases compared to both the A and M groups.

To elucidate the molecular mechanisms underlying the observed changing muscle fiber type composition, the focus was placed on genes related to muscle fiber type switching, particularly those associated with slow-twitch fibers, which are crucial for muscle endurance (myocyte enhancer factor 2C: MEF2C). Analysis of MEF2C gene expression revealed a significant upregulation in all groups (A, M, AML, and AMH), except for the PC group, compared to the NC group (Figure 5c). Notably, the AML group exhibited a significantly higher expression level than the A and M groups, suggesting there is a synergistic effect between two extracts (Figure 5c).

## 4. Discussion

This study investigates a novel candidate that can significantly enhance muscle strength and endurance when combined with mistletoe extract. While mistletoe extract alone demonstrates efficacy in enhancing muscle strength and muscular endurance at a high dose [47], safety and economic limitations necessitate the exploration of alternatives. We selected apple peel extract as a candidate and aimed to comprehensively evaluate the effects of apple peel extract alone, mistletoe extract alone, and their combination on muscle strength and endurance, compared to the control groups. Prior to conducting *in vivo* studies, the total polyphenol content and the flavonoid portion in the samples were quantified for use in all experimental groups. Although both mistletoe and apple peel extracts contain similar amounts of polyphenols per unit volume, the mistletoe extract has significantly fewer flavonoids compared to the apple peel extract (Figure 1c). This suggests that most of the polyphenols in the mistletoe extract are non-flavonoids, such as phenolic acids, a major class within the non-flavonoids [48]. 

The results of the animal behavior experiments surpassed the projected outcomes. Administering a combination of mistletoe extract and apple peel extract, each at a dose of 100 mg/kg (AML group: total dose of 200 mg/kg), resulted in a significant increase in muscle strength and endurance compared to the groups receiving either extract individually at 200 mg/kg (A group and M group) (Figure 2b,c). These observations suggest a synergistic effect between mistletoe and apple peel extracts in enhancing muscle strength and muscular endurance. The potential mechanisms underlying the observed synergy were inferred by analyzing key biomarkers related to muscle strength and endurance.

The initial investigation focused on the morphological characteristics of the gastrocnemius (GA) muscles located in the back part of the lower leg. It is involved in movements such as walking, running, and jumping [49]. While no significant differences were observed in calorie intake, body weight, and muscle weight among the test groups (Figure 2d–f), the AML and AMH groups displayed larger cross-sectional areas (CSAs) of GA muscle fibers compared to the A and M groups (Figure 3c). These results showed that the increase in the CSA is due to the co-administration of mistletoe and apple peel extracts, rather than an increase in calorie intake. Furthermore, the significant reduction in the non-muscle fiber area within the GA muscle explains the increase in the muscle fiber area without changes in overall muscle mass (Figure 3b). This muscle fiber enlargement indicates an increase in contractile proteins like myosin and actin within the muscle tissue, which is likely involved in boosting muscle strength [50].

To explore the molecular mechanisms underlying the synergistic effect on muscle strength, changes in gene expression levels related to myogenesis, protein synthesis, and protein degradation in the GA muscles were investigated. Specifically, for AKT, which is known to play key roles in enhancing muscle strength, changes in both RNA and protein levels were measured (Figure 3h–j). In general, genes involved in protein degradation (FOXO1, MuRF1, Atrogin-1, TNFα, and NF-κB) showed downregulation, while genes related to muscle formation (MyoD) and protein synthesis (IGF1, S6K, and AKT) were upregulated in all test groups compared to the NC group, except in the M group (Figure 3e–h). Interestingly, despite using 50% less mistletoe extract than the M group, the AML group, when co-administered with the same amount of apple peel extract (A: 100 mg/kg + M: 100 mg/kg), showed a greater increase in the expression of genes related to muscle protein synthesis—S6K (protein synthesis), AKT (protein synthesis and degradation), and a great decrease in the MuRF1 (protein degradation) expression—compared to the groups that received the individual extracts alone at 200 mg/kg (A group or M group) (Figure 3e,g–h). Additionally, in the AML and AMH groups, the RNA levels of Atrogin-1, TNFα, NF-κB, MyoD, and IGF1 changed in a dose-dependent manner (Figure 3e-g), indicating that this mixed extract contains active components that synergistically regulate gene expression related to muscle strength.

The densities of mitochondria in GA muscles were then measured using transmission electron microscopy (TEM) (Figure 4a). Mitochondrial density did not significantly increase in the A group but did significantly increase in the M group compared to the NC group (Figure 4b). This result is consistent with previous discoveries that mistletoe extract can enhance mitochondrial biogenesis [51]. However, the AML group exhibited an even higher mitochondrial density than the M group (Figure 4b). This result is consistent with the synergistic increase in running time observed in the AML group (Figure 2c; more than 2× compared to the NC/PC/A groups and more than 1.5× compared to the M group). Increased mitochondrial biogenesis boosts ATP production via oxidative phosphorylation, improving energy efficiency [52]. Additionally, increased mitochondrial biogenesis dilutes ROS, reducing oxidative stress and enhancing cell health [53]. Molecular markers associated with mitochondrial biogenesis also support these findings (Figure 4e–g). Co-administration of mistletoe and apple peel extracts synergistically increased the expression of key regulators of mitochondrial biogenesis, including PGC-1α and TFAM (Figure 4e).

Mitochondrial dynamics, involving fission and fusion, also play a key role in enhancing muscle endurance [54]. While mitochondrial fission removes severely damaged mitochondria (preventing the accumulation of dysfunctional organelles and handling acute damage), fusion promotes the repair of mildly damaged mitochondria, optimizing their energy production and helping them recover from and adapt to prolonged stress [55,56]. Notably, the mitochondrial fission marker levels were particularly low in the group treated with apple peel extract alone (A group; Figure 4c). These levels were as low as those in the group that received both apple peel and mistletoe extracts in equal amounts (AMH group). These results imply that the mitochondria are functioning well and maintaining their integrity at a high level in the A and AMH groups. The abundant antioxidants in apple peel extract may have contributed to protecting the mitochondria and maintaining their overall health [57]. In the M group, which was administered mistletoe extract alone, there was a significant decrease in the fission markers compared to the NC group, although not as pronounced as in the A group. This is likely related to the high biogenesis activity indicators, such as SIRT and TFAM, observed in the M group. Generally, when mitochondrial biogenesis is active, the proportion of newly generated healthy mitochondria increases, leading to a decrease in fission-related indicators [58]. For mitochondrial fusion markers, similar to the fission markers, both the AMH group and the A group showed the most impressive increase in expression (Figure 4d). It is known that increased fusion activity enhances the sustainability of mitochondrial function [59].

Muscular endurance is also influenced by the types of muscle fiber present [60]. These fibers are classified based on their contractile properties, metabolic characteristics, and resistance to fatigue [61,62]. Type I fibers, also known as slow-twitch fibers, are rich in mitochondria and contain significant amounts of myoglobin, enabling efficient oxygen use, which makes them well-suited for long-duration activities [63]. Type IIa fibers, while having a faster contraction speed than Type I fibers, also possess moderate endurance [64]. In contrast, Type IIb fibers fatigue more quickly and have lower endurance than other fiber types but possess the fastest contraction speed and highest force output, making them ideal for short bursts of activity [65]. Mistletoe extract (M) significantly increases Type I and Type IIa fiber markers compared to both the control groups (NC and PC) and the A group (Figure 5b). In the AML group, the increase in Type I and Type IIa markers is even greater than when either extract is used alone. For the Type IIb markers, the AML group also showed an increase compared to the A or M groups, but the statistical significance was lower than that observed for the Type I and Type IIa fiber markers.

How can the synergistic increase in both muscle strength and especially in muscle endurance be explained at the molecular level? Analysis of the signaling network (Figure 6) suggests that the observed synergy might be due to the simultaneous occurrence of a decrease in myostatin (MSTN gene product) induced by mistletoe extract and an increase in IGF1 (Insulin-like Growth Factor 1) induced by apple peel extract in the AML and AMH groups. IGF1 and myostatin, two small proteins, exert opposing effects on myogenesis [66]. IGF1 stimulates muscle growth by activating the PI3K/Akt/mTOR pathway, whereas myostatin inhibits muscle growth through the activation of the SMAD2/3 signaling pathway. Additionally, myostatin can antagonize the anabolic effects of IGF1 by suppressing the PI3K/Akt/mTOR pathway via inhibition of Akt activity. Both proteins also play significant roles in regulating mitochondrial biogenesis [67]. The principal regulator of mitochondrial biogenesis is PGC-1α [68], which IGF1 can activate through mTORC1 within the PI3K/Akt/mTOR pathway. In contrast, myostatin downregulates PGC-1α via the SMAD2/3 signaling pathway [69]. When mistletoe extract was administered alone (M group), the expression level of MSTN significantly decreased, but there was no significant change in IGF1 expression (Figure 6b). In contrast, when apple peel extract was administered alone (A group), there was a significant increase in IGF1 expression, but no significant change in MSTN expression (Figure 6a). When both extracts were administered together, significant changes in both MSTN and IGF1 expression were observed (Figure 6c). These changes likely contributed to the synergistic improvement in muscle strength by simultaneously relieving the inhibition of PI3K activity by the decreased levels of myostatin and enhancing PI3K activation by increased levels of IGF1 in the PI3K-AKT-mTOR pathway, which is crucial for muscle growth and protein synthesis. The markedly increased phosphorylation of AKT, a downstream protein of PI3K, in the AML and AMH groups aligns well with this synergistic effect (Figure 3i,j). The phosphorylated AKT upregulated a downstream gene S6K significantly (Figure 3g and Figure 6c), which can enhance muscle protein synthesis [70]. MuRF1 is another key gene involved in the degradation of proteins such as myosin and actin [71]. FOXO1 and NF-κB are transcription factors that bind to the promoter of MuRF1, stimulating its expression [72,73]. The activated AKT also repressed the expression of both FOXO1 and NF-κB, leading to a synergistic reduction in the expression of MuRF1 (Figure 3e and Figure 6c), which in turn decreases muscle protein degradation [74].

Administration of both extracts also significantly enhances muscle endurance. Notably, the interaction of the combined extracts highly activates the PI3K/Akt/mTOR pathway, and particular attention should be paid to the significantly increased AKT phosphorylation during this process (Figure 3j). In the M group, the level of AKT phosphorylation was not significantly different from that of the negative control group. However, in the AML group, it showed a substantial increase, likely due to the increase in IGF1. Phosphorylated AKT in the AML group enhances mTOR expression through the PI3K-AKT-mTOR pathway, and mTOR can increase PGC-1α expression by activating the transcription factor Yin Yang 1 (YY1) [75]. In the AML and AMH groups, the marked increase in PGC1α RNA expression observed (Figure 4e) provides evidence supporting this finding. PGC-1α is not a kinase; it is a transcriptional coactivator that regulates the genes involved in energy metabolism and mitochondrial biogenesis [76]. It does not bind DNA directly but interacts with various transcription factors. This interaction enhances the transcription factors’ affinity for DNA, recruits additional coactivators, and stabilizes the transcriptional complex [77]. For example, PGC-1α interacts with NRF-1 and NRF-2 transcription factors to promote the transcription of TFAM, which plays a crucial role in mitochondrial biogenesis [78]. PGC-1α can also interact with MEF2C, which directly enhances the transcription of the MYH7 gene, essential for the slow and sustained contraction properties of type I fibers [79]. These findings align with this study’s results, which showed that TFAM and MEF2C transcription increased significantly with elevated PGC-1α expression (Figure 4e and Figure 5c). This not only enhances the number and function of mitochondria within the muscle but also alters muscle fiber type composition by increasing Type I and Type IIa fibers, thereby contributing to improved muscle endurance performance.

What are the active compounds in the mixed extract contributing to the observed synergy? This study used two natural extracts, making it difficult to pinpoint which compounds were responsible for the observed effects. However, the polyphenol composition analysis showed that mistletoe extract contains significantly fewer flavonoids compared to apple peel extract (Figure 1c). This is supported by other studies, which quantified total phenolic, flavonoid, and hydroxycinnamic acid contents in various mistletoe types. They found that only approximately 10% of total phenolics in mistletoe are flavonoids [80], whereas in apples, over 50% of the total phenolic content is flavonoids [34]. Mistletoe contains a rich diversity of non-flavonoid compounds with various bioactive properties. The phenolic acids present in mistletoe, such as chlorogenic acid and caffeic acid, have notable effects on improving muscle strength and muscular endurance. Chlorogenic acid significantly improves muscle strength and mitochondrial function [81]. Caffeic acid blocks NF-κB activation and reduces prooxidant and proinflammatory responses, significantly suppressing inflammation-related genes (IL-β, MCP-1, iNOS, COX-2) and oxidative stress marker MDA [82]. 

On the other hand, apple peel is rich in flavonoids such as quercetin and its glycosides [83]. Quercetin has been shown to boost AKT phosphorylation [84,85,86]. Additionally, quercetin treatment reduces MuRF1 and Atrogin-1 expression and promotes myotube growth by activating the AKT/mTOR/S6K pathway [87,88]. Furthermore, both mistletoe and apple contain triterpenic acids such as oleanolic, ursolic, and betulinic acids, which various studies have demonstrated to be effective in enhancing muscular strength and endurance [89,90,91,92]. Further research is needed to determine whether or not the abundant flavonoids in apples contribute to the synergistic effects observed between these two extracts.

In this study, a significant synergistic effect between the two substances was observed following a short 4-week oral administration. However, several limitations were identified. While a dose-dependent synergistic effect was confirmed in many molecular markers related to muscle strength and endurance between the AML and AMH groups, the grip test and treadmill test did not show statistically significant dose-dependent synergy between them. A longer administration period beyond 4 weeks may be necessary to achieve statistically significant results, as previous research in metabolic disease models suggests that treatment durations of 8 to 12 weeks are required to detect changes in markers such as body weight and insulin [93]. Another possible explanation is that the 100 mg/kg dose of each extract may have reached a plateau in its effect on muscle strength. This plateau effect, known as the ceiling effect, occurs when increasing doses fail to produce additional effects, potentially due to receptor saturation [94,95,96]. This suggests a sigmoidal dose-response relationship with respect to muscle strength and endurance.

Moreover, how these interactions led to such levels of improvement and which gene expressions or interactions exerted the most significant influence on the results are yet to be fully elucidated. The regulation of muscle strength and endurance is highly complex, involving processes such as muscle formation, regulation, degradation, and the influence of factors such as ATP supply, insulin concentration in the blood, and inflammation in the muscle [97]. To further elucidate these intricate mechanisms, future research will integrate both computational modeling and in vivo studies. This combined approach will facilitate a more comprehensive analysis of the complex interactions within biological systems, particularly by revealing how changes in one component of the network can impact the system as a whole [98,99,100].

Finally, while synergistic enhancements were demonstrated in the mouse model, caution is necessary when extrapolating these findings to humans. Differences in metabolic processes, physiology, and dosage responses between species may affect the effectiveness of these extracts in human applications. However, both mistletoe and apple peel contain bioactive compounds, such as phenolic acids and flavonoids, which have been shown to positively impact muscle function and health in various human studies. Future clinical trials are needed to confirm the efficacy, optimal dosages, and safety of these extracts in human populations. Such trials could provide valuable insights into whether or not the synergistic effects observed in animals can be replicated in humans, potentially paving the way for novel, natural interventions to support muscle health.

## 5. Conclusions

This study demonstrates that the combination of Korean mistletoe and apple peel extracts exerts significant synergistic effects on muscle strength and endurance in a mouse model. The combination enhanced muscle protein synthesis, reduced muscle degradation, promoted mitochondrial biogenesis and function, and altered muscle fiber type distribution, surpassing the effects observed with individual extract treatments. However, further research, including human clinical trials, is needed to confirm the efficacy and safety of these extracts in human populations, as well as to determine optimal dosing strategies for maximizing benefits. The insights gained from this study open the door to the potential development of natural supplements aimed at improving muscle health.

## 6. Patents

Patent applications related to this work have been filed by Jrapha Co., Ltd. (Korea Patent application number 10-2024-0024535).

## Figures and Tables

**Figure 1 nutrients-16-03255-f001:**
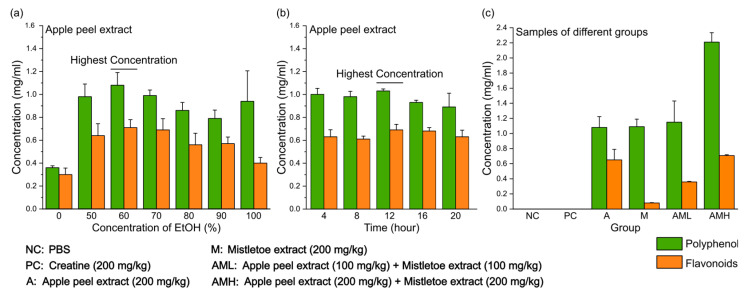
Polyphenol and flavonoid concentrations in apple peel extracts under different extraction conditions. (**a**) The yield of polyphenol and flavonoids at different ethanol concentrations. (**b**) The effect of varying extraction times on polyphenol and flavonoids. (**c**) Polyphenol and flavonoids concentrations in each sample for oral administration. Results are presented as means ± S.D. (*n* = 6).

**Figure 2 nutrients-16-03255-f002:**
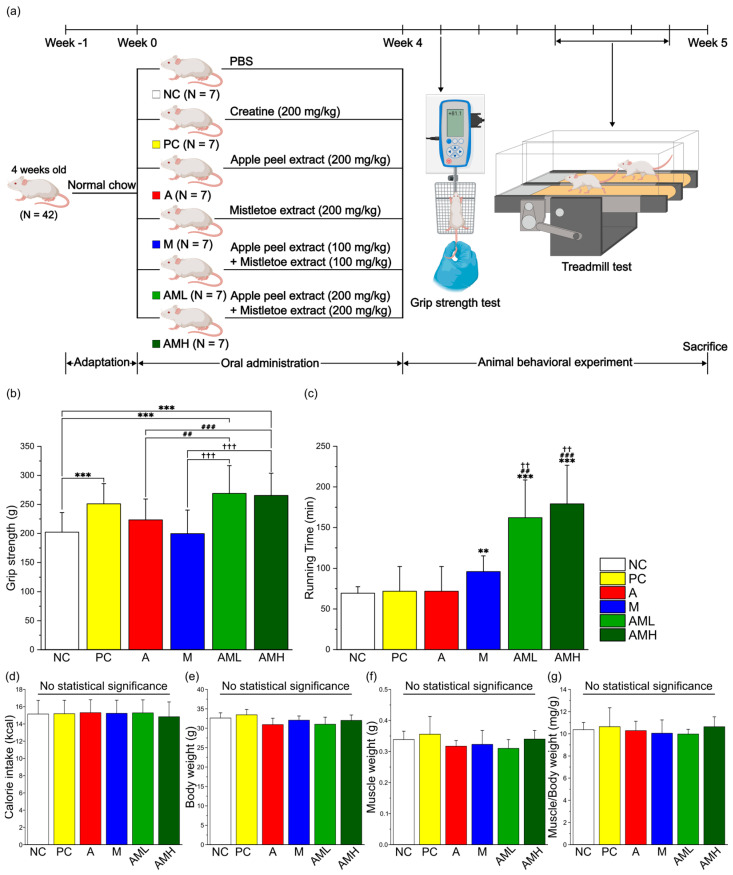
Combination of mistletoe extract and apple peel extract has a synergistic effect on enhancing muscle strength and muscular endurance. (**a**) Schematic diagram of the animal experiment. (**b**) Muscle strength measured by grip strength meter. (**c**) Running time during treadmill test. (**d**) Average of calorie intake per day. (**e**) Changes in body weight during the animal experiment. (**f**) Gastrocnemius (GA) muscle weight after 4 weeks of treatment. (**g**) Ratio of GA weight to body weight after 4 weeks of treatment. Results are presented as means ± S.D (*n* = 7). Significance level is indicated as ** *p* < 0.01, *** *p* < 0.001 for comparison with the NC group; ## *p* < 0.01, ### *p* < 0.001 for comparison with A group; and †† *p* < 0.01, ††† *p* < 0.001 for comparison with M group. NC: PBS, PC: Creatine (200 mg/kg), A: Apple peel extract (200 mg/kg), M: Mistletoe extract (200 mg/kg), AML: Apple peel extract (100 mg/kg) and mistletoe extract (100 mg/kg), AMH: Apple peel extract (200 mg/kg) and mistletoe extract (200 mg/kg).

**Figure 3 nutrients-16-03255-f003:**
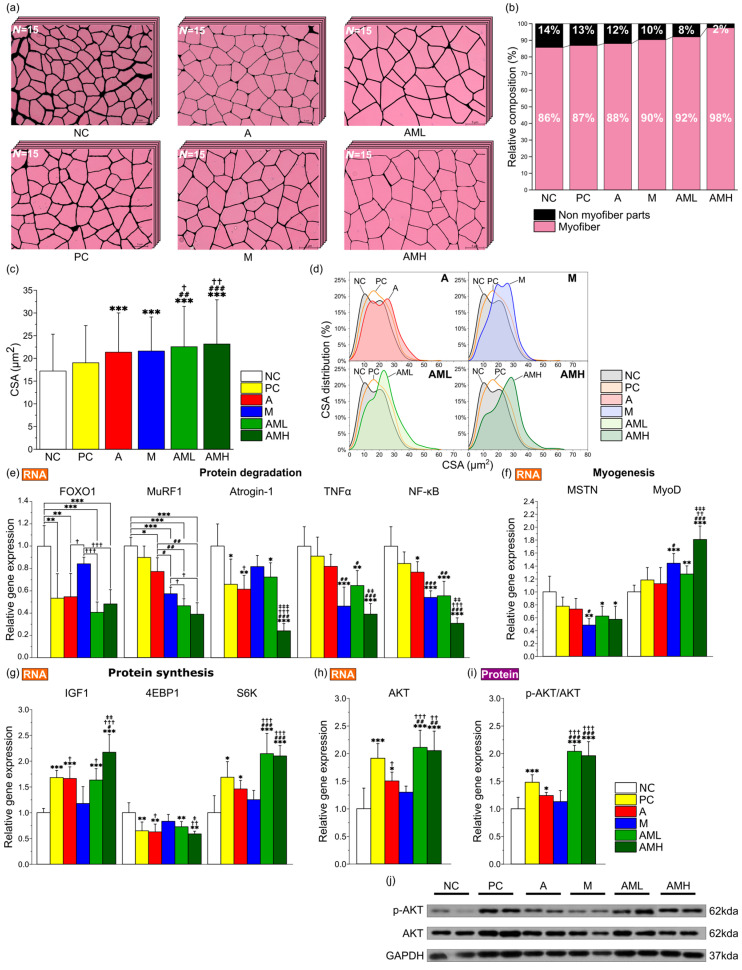
Combination of mistletoe extract and apple peel extract exerts synergistic effect on increasing size of muscle fibers. (**a**) Representative merged images of GA muscle tissue sections stained with H&E at 200× magnification. Individual muscle fibers are distinguished and labeled using ImageJ software (*n* = 15). (**b**) Percentage of proportion of myofiber and connective tissue layer. (**c**) Average CSA of muscle fibers. (**d**) Distribution of muscle fiber CSA. Relative expression levels of genes related to (**e**) protein degradation (FOXO1, MuRF1, Atrogin-1, TNFα, and NF-κB), (**f**) myogenesis (MSTN and MyoD), (**g**) Protein synthesis (IGF-1, 4EBP1, and S6K), and (**h**) AKT in GA muscle tissue (*n* = 7). (**i**) Ratio of p-AKT to total AKT protein expression in GA muscle tissue. (**j**) Representative Western blot images of protein expressions. CSA was measured using ImageJ software. Results are presented as means ± S.D. Significance level are indicated as * *p* < 0.05, ** *p* < 0.01, *** *p* < 0.001 for comparison with the NC group; # *p* < 0.05, ## *p* < 0.01, ### *p* < 0.001 for comparison with A group; † *p* < 0.05, †† *p* < 0.01, ††† *p* < 0.001 for comparison with M group; and ‡ *p* < 0.05, ‡‡ *p* < 0.01, ‡‡‡ *p* < 0.001 for comparison between AML and AMH groups.

**Figure 4 nutrients-16-03255-f004:**
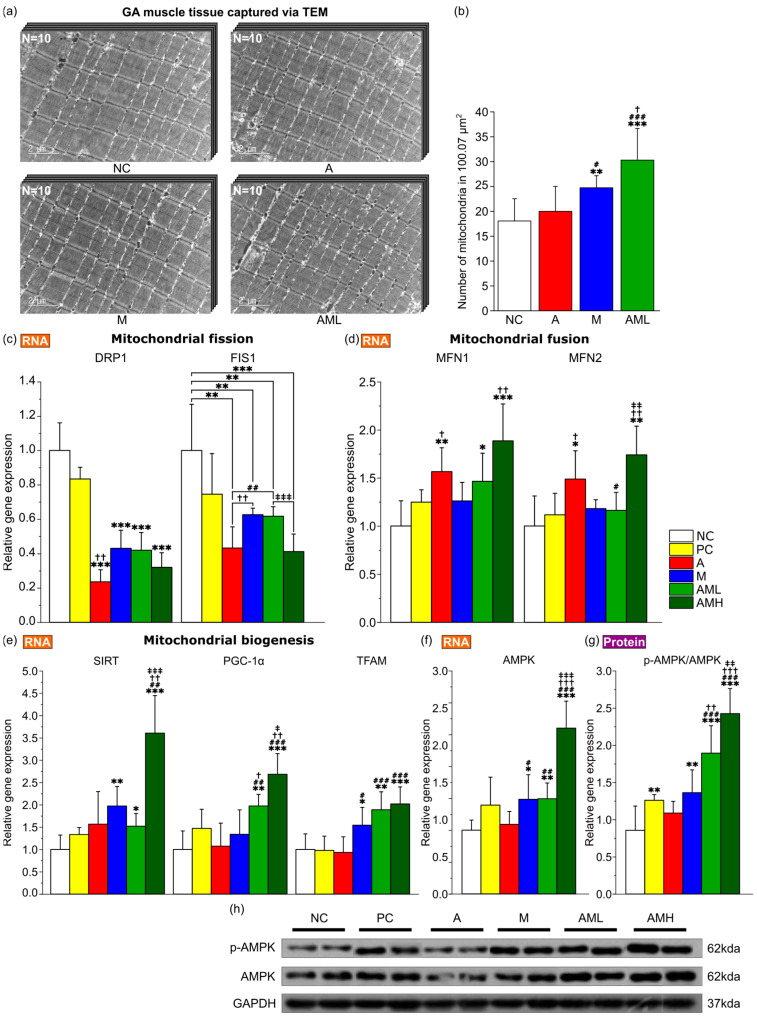
Impact of the combination of mistletoe extract and apple peel extract on mitochondrial biogenesis and dynamics. (**a**) Transmission electron microscopy images of GA muscle tissue at 15000× magnification. Scale bars: 2 µm. (**b**) Number of mitochondria in GA muscle tissue (*n* = 10). Relative expression of genes related to (**c**) mitochondrial fission (DRP1 and FIS1), (**d**) mitochondrial fusion (MFN1 and MFN2), (**e**) mitochondrial biogenesis (SIRT1, PGC-1α, and TFAM) and (**f**) AMPK in GA muscle tissue (*n* = 7). (**g**) The ratio of p-AMPK to total AMPK protein expression in GA muscle tissue. (**h**) Representative Western blot images of protein expressions. Results are presented as means ± S.D. Significance level are indicated as * *p* < 0.05, ** *p* < 0.01, *** *p* < 0.001 for comparison with the NC group; # *p* < 0.05, ## *p* < 0.01, ### *p* < 0.001 for comparison with A group; † *p* < 0.05, †† *p* < 0.01, ††† *p* < 0.001 for comparison with M group; and ‡ *p* < 0.05, ‡‡ *p* < 0.01, ‡‡‡ *p* < 0.001 for comparison between AML and AMH groups.

**Figure 5 nutrients-16-03255-f005:**
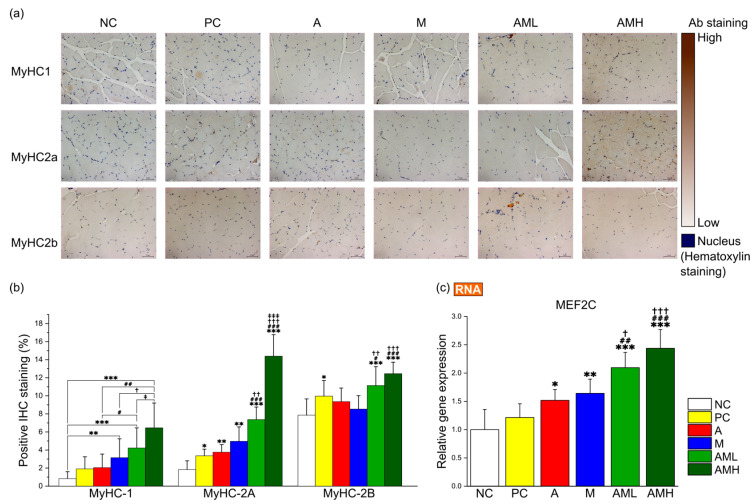
Combination of mistletoe extract and apple peel extract synergistic effect on fiber type switching in gastrocnemius muscle tissue. (**a**) Representative images of IHC for the GA muscle at 200× magnification. (**b**) Proportion of MyHC isoform-specific fibers. (**c**) Relative gene expression of MEF2C. The percentage of stained area by IHC was calculated using ImageJ. Results are presented as means ± S.D (*n* = 5). Significance levels are indicated as * *p* < 0.05, ** *p* < 0.01, *** *p* < 0.001 for comparison with the NC group; # *p* < 0.05, ## *p* < 0.01, ### *p* < 0.001 for comparison with A group; † *p* < 0.05, †† *p* < 0.01, ††† *p* < 0.001 for comparison with M group; and ‡ *p* < 0.05, ‡‡‡ *p* < 0.001 for comparison between AML and AMH groups.

**Figure 6 nutrients-16-03255-f006:**
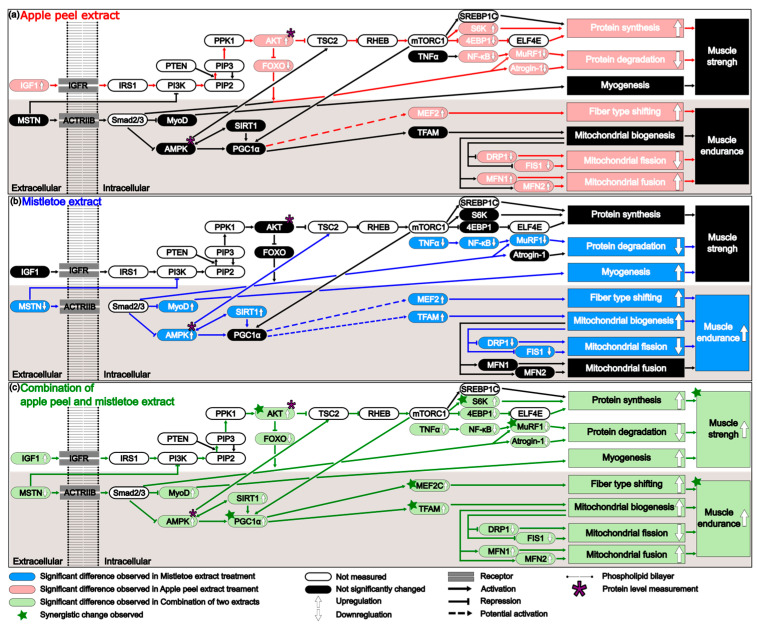
The mechanism underlying the effects of each extract on muscle strength and muscular endurance when administered individually or in combination. (**a**) Apple peel extract enhances muscle strength by increasing protein synthesis and decreasing protein degradation. It also enhances muscular endurance by increasing muscle fiber type shifting, increasing mitochondrial fusion, and decreasing mitochondrial fission. (**b**) Mistletoe extract enhances muscle strength by reducing protein degradation and promoting myogenesis. It also improves muscular endurance by shifting muscle fiber types, increasing mitochondrial biogenesis, and decreasing mitochondrial fission. (**c**) The combination of apple peel and mistletoe extracts enhance muscle strength and muscular endurance. The genes marked with a green star showed statistically significant changes when administered a combination of two extracts, compared to when each extract was administered alone. The genes marked with a purple asterisk (AMPK and AKT) were measured not only at the RNA level but also at the protein level.

**Table 1 nutrients-16-03255-t001:** All groups of animal experiments.

Groups	Treatment
NC	Negative control: 200 µL 1 × phosphate buffered saline (PBS)
PC	Positive control: 200 mg/kg creatine
A	200 mg/kg apple peel extract
M	200 mg/kg mistletoe extract
AML	Low dose: 100 mg/kg apple peel extract + 100 mg/kg mistletoe extract
AMH	High dose: 200 mg/kg apple peel extract + 200 mg/kg mistletoe extract

## Data Availability

The original contributions presented in the study are included in the article/Appendix A; further inquiries can be directed to the corresponding author/s.

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
