# Peer review of "Synergistic Effects of Korean Mistletoe and Apple Peel Extracts on Muscle Strength and Endurance"

_nutrients, 2024, doi:10.3390/nu16193255_

Round 1

Reviewer 1 Report

Comments and Suggestions for Authors

This is a high-quality study analyzing synergistic effect of Korean mistletoe and apple peel extracts on muscle strength and endurance in mice. The article is characterized by extraordinary clarity. The introduction perfectly introduces the reader to the subject of the work, justifying the undertaken research. Materials and methods are meticulously described. The results of this complex research are logically described and presented in graphs and then subjected to an in-depth discussion. It is rare to see such a well-thought-out research work, so clearly presented and thoroughly analyzed from the perspective of scientific analysis.

I have only two remarks. First, I believe that it is necessary to add conclusion chapter in Nutrients. Second, PBS abbreviation in Table 1. should be explained there, not in the line 247.

Author Response

Comments1: Add conclusion chapter in Nutrients.
Response 1: Thank you for pointing this out. I agree with this comment. Therefore, I add the conclusion chapter in the manuscript. The conclusion chapter is composed of lines 685 to 694.

Comments 2: PBS abbreviation in Table 1. should be explained there, not in the line 247.
Response 2: Thank you for pointing this out. I agree with this comment. I added the PBS abbreviation in Table 1 not in the line 247.

Reviewer 2 Report

Comments and Suggestions for Authors

Well written manuscript, interesting topic; the synergistic finding is interesting?  Does this suggest what should be done for a supplement?

L24: I think a more recent citation than 1985 is warranted.  Anything from the American College of Sports Medicine?

L26: is body composition an indicator of fitness?

L47: with a statement like "Throughout history..."  that there should be more than just one reference to support it

L51: grammar revision suggestiong: The administration of 40mg/kg/day of reserveratrol to rodents demonstrated a significant.....

Was there a control group with that study?  Were either group training or were they restricted from activity?

L54: instead of "and" use "versus"

L70: Avoid 1st person.  So don't use "we".  Make the change to 3rd person.  I will highlight other "we" to address (there may be more)

L78: We

Well written methodology, great job quantifying each step of the procedure

Excellent use of figures

L315: We

L423: We

L435: We

L457: We

L467: Avoid conversational language.  Should not use "exceeded our expectations".  

L504: We

L657: We

Author Response

Comments 1: L24: Need more recent citation than 1985 is warranted. 

Response 1: Thank you for pointing this out. I agree with this comment. In line 28, the 2022 paper by Gupta et al. has been changed as a reference.
Gupta, S. R. (2022). A comprehensive study on the relevance of health, fitness, and wellness. Journal of Sports Science and Nutrition, 3(1), 145-148.

Comments 2: L26: Is body composition an indicator of physical fitness?

Response 2: Thank you for pointing this out. As mentioned in the manuscript, body composition is one of the components of physical fitness. Additional related papers are provided as references (Nuzzo. 2020) in Line 30.
Nuzzo, J. L. (2020). The case for retiring flexibility as a major component of physical fitness. Sports Medicine, 50(5), 853-870.

Comments 3: L47: with a statement like "Throughout history..."  that there should be more than just one reference to support it

Response 3: Thank you for pointing this out. I agree with this comment. In Line 51, two additional papers have been added as references to support this statement (Yu et al. 2023, Kuvačić et al. 2018)

Yu, Z., Wang, W., Yang, K., Gou, J., Jiang, Y., & Yu, Z. (2023). Sports and Chinese herbal medicine. Pharmacological Research-Modern Chinese Medicine, 100290.Sellami, M., Slimeni, O., Pokrywka, A., Kuvačić, G., D Hayes, L., Milic, M., & Padulo, J. (2018). Herbal medicine for sports: a review. Journal of the International Society of Sports Nutrition, 15, 1-14.

Comments 4: L51: grammar revision suggestiong: The administration of 40mg/kg/day of reserveratrol to rodents demonstrated a significant.....

Response 4: Thank you for pointing this out. I agree with this comment. In line 54 - 56, the sentence has been revised to: A rodent study administering resveratrol at a dose of 400 mg/kg per day showed a significant reduction in muscle fiber atrophy.

Comments 5: Was there a control group with that study?  Were either group training or were they restricted from activity?

Response 5: This study utilized a total of four experimental groups: a control group with no intervention (C), a hypokinetic and hypodynamic hindlimb suspended group (HH), a group receiving only resveratrol (CR), and a group that was both hindlimb suspended and given resveratrol (HHR). First, the groups were divided based on whether gravity was applied or minimized. Then, each group was further divided based on whether resveratrol was administered or not, allowing for an examination of the effects of resveratrol on muscle.

Comments 6: L54: change the word from "and" to "versus"

Response 6: Thank you for pointing this out. I agree with this comment.

In line 57, the sentence has been revised to: A study comparing the effects of mistletoe extracts versus resveratrol on exercise performance in mice found that mistletoe extracts resulted in significantly longer swimming times (over 500 mins) compared to resveratrol (approximately 400 mins) and the negative control (approximately 190 mins).

Comments 7: Avoid 1st person. So don't use "we".  Make the change to 3rd person.

Response 7: Thank you for pointing this out. I agree with this comment. All first-person sentences beginning with 'We' have been revised.

Line 73: To address the limitations of mistletoe, this study explored natural products that could be combined with mistletoe extract to complement its limitations while enhancing its effectiveness.

L81: We hypothesized that combining mistletoe with apple extracts could potentially enhance muscle strength and endurance due to the complementary effects of their different compositions including polyphenolic compounds.

We agree with the reviewer's comment and have removed 'we' from all the other sentences pointed out. However, in this particular case, it is necessary to emphasize that the hypothesis was made by our research team, and since the use of 'we' is allowed in a limited way when stating the study's hypothesis or conclusions, we have used 'we' in this sentence.

L236: To effectively extract polyphenols, including flavonoids, which are considered one of the most beneficial compounds in apple peels, the effects of varying ethanol concentrations and extraction time in ethanol were compared.

L317: The analysis focused on genes related to protein degradation (Forkhead box protein O1; FOXO1, Muscle RING-finger protein-1; MuRF1, F-box protein 32; Atrogin-1, Tumor necrosis factor alpha; TNFα, and Nuclear factor kappa B; NF-κB), myogenesis (Myostatin; MSTN, Myoblast determination protein 1; MyoD), and protein synthesis (Insulin-like growth factor-1; IGF1, Eukaryotic translation initiation factor 4E-binding protein 1; 4EBP1, Ribosomal protein S6 kinase; S6K, and Protein kinase B; AKT). 

L323: Additionally, the phosphorylation level of AKT at Ser473, a critical marker of protein activation, was observed

L377: To elucidate the molecular mechanisms underlying the observed changes in muscular endurance, The focus was on genes related to mitochondrial biogenesis (Sirtuin 1; SIRT1, Peroxisome proliferator activated receptor gamma coactivator 1 alpha; PGC-1ɑ, and Transcription factor A, mitochondrial; TFAM), mitochondrial fission (Dynamin related protein; DRP1 and mitochondrial fission 1 protein; FIS1), mitochondrial fusion (mitofusin 1; MFN1 and mitofusin 2; MFN2) and AMP activated protein kinase (AMPK) along with AMPK phosphorylation as a marker of its relevant pathway activation.

L425: Based on these results, immunohistochemistry (IHC) was employed on GA muscle cross-sections to determine the effect of this combination on muscle fiber type composition.

L427: The antibodies targeting specific markers for type I (MyHC-1), type IIa (MyHC-2A), and type IIb (MyHC-2B) fibers were utilized (Figure 5a). 

L 437: To elucidate the molecular mechanisms underlying the observed changing muscle fiber type composition, the focus was placed on genes related to muscle fiber type switching, particularly those associated with slow-twitch fiber, which are crucial for muscle endurance (Myocyte enhancer factor 2C; MEF2C)

L 460: We selected apple peel extract as a candidate and aimed to comprehensively evaluate the effects of apple peel extract alone, mistletoe extract alone, and their combination on muscle strength and endurance, compared to the control groups.

In line with the earlier description of the hypothesis, it was hoped that 'we' would be used in this section as well, and thus 'we' was only used in these two parts, with all other sections modified according to the reviewer's advice.

L 463: Prior to conducting in vivo studies, the total polyphenol content and the flavonoid portion in the samples were quantified for use in all experimental groups.

L 476: The potential mechanisms underlying the observed synergy were inferred by analyzing key biomarkers related to muscle strength and endurance.

L 479: The initial investigation focused on the morphological characteristics of the gastrocnemius (GA) muscles located in the back part of the lower leg.

L 491: To explore the molecular mechanisms underlying the synergistic effect on muscle strength, changes in gene expression levels related to myogenesis, protein synthesis and protein degradation in the GA muscles were investigated.

L 493: Specifically, for AKT, which is known to play key roles in enhancing muscle strength, changes in both RNA and protein levels were measured (Figure 3h-j).

L 509: The densities of mitochondria in GA muscles were then measured using transmission electron microscopy (TEM) (Figure 4a).

L 627: This study used two natural extracts, making it difficult to pinpoint which compounds were responsible for the observed effects. 

L 663 : While synergistic effects on muscle strength and endurance were demonstrated in this study, and specific genes and their interactions related to these effects were identi-fied, how these interactions led to such levels of improvement, and which genes or in-teractions have the greatest impact on the results, remain unclear. 

Comments 8: Avoid conversational language. Should not use "exceeded our expectations".

Response 8: Thank you for pointing this out. In line 470, the sentence has been revised to: The results of the animal behavior experiments surpassed the projected outcomes.

Reviewer 3 Report

Comments and Suggestions for Authors

The study conducted by Kang and collaborators is relevant, and original and gives the audience pertinent results. I believe it can be considered for publication after the following revisions:

You should provide the main values of your results in the abstract. It would help if you also gave the conclusions, practical implications, and directions for future studies.

The whole manuscript is well-written and clear. The background provided in the Introduction section and the methodologies used seem adequate. The results are properly demonstrated and discussed with updated references.

My question to the authors is about the extrapolation of the obtained results to humans. I encourage the authors to elaborate on this in the manuscript and to improve the conclusions on this basis. In my point of view, conclusions should be included in a separate section.

Author Response

Comments 1:  provide the main values of your results in the abstract. It would help if you also gave the conclusions, practical implications, and directions for future studies.

Response 1: Thank you for pointing this out. I have added more details to the abstract based on the comments provided. (Line 20 to Line 23) This research provides practical insights into developing cost-effective, natural supplements to enhance muscle performance and endurance, with potential applications in athletic performance, improving muscle growth and endurance in children, and addressing age-related muscle decline.

Comments 2:  The authors are about the extrapolation of the obtained results to humans.

Response 2: Thank you for pointing this out. I added the extrapolation of the obtained results to humans.

(Line 674 to Line 684) Although significant synergistic effects of apple peel and mistletoe extracts on muscle strength and endurance were observed in a mouse model, caution is necessary when extrapolating these findings to humans. Differences in metabolic processes, physiology, and dosage responses between species may affect the effectiveness of these extracts in human applications. However, both mistletoe and apple peel contain bioac-tive compounds, such as phenolic acids and flavonoids, which have been shown to positively impact muscle function and health in various human studies. Future clinical trials are needed to confirm the efficacy, optimal dosages, and safety of these extracts in human populations. Such trials could provide valuable insights into whether the synergistic effects observed in animals can be replicated in humans, potentially paving the way for novel, natural interventions to support muscle health.

Comments 3:  The authors are about the extrapolation of the obtained results to humans.

Response 3: Thank you for pointing this out. Therefore, I add the conclusion chapter in the manuscript. The conclusion chapter is composed of lines 685 to 694.